# Promotion of E-Cigarettes on TikTok and Regulatory Considerations

**DOI:** 10.3390/ijerph20105761

**Published:** 2023-05-09

**Authors:** Jonine Jancey, Tama Leaver, Katharina Wolf, Becky Freeman, Kevin Chai, Stella Bialous, Marilyn Bromberg, Phoebe Adams, Meghan Mcleod, Renee N. Carey, Kahlia McCausland

**Affiliations:** 1Collaboration for Evidence, Research and Impact in Public Health, School of Population Health, Curtin University, Kent Street, Perth, WA 6102, Australia; 2Internet Studies, School of Media, Creative Arts and Social Inquiry, Curtin University, Kent Street, Perth, WA 6102, Australia; 3School of Marketing, Curtin University, Kent Street, Perth, WA 6102, Australia; 4School of Public Health, University of Sydney, Sydney, NSW 2006, Australia; 5School of Population Health, Curtin University, Kent Street, Perth, WA 6102, Australia; 6School of Nursing, University of California, San Francisco, CA 94158, USA; 7UWA Law School, The University of Western Australia, 35 Stirling Highway, Perth, WA 6009, Australia

**Keywords:** e-cigarettes, public health, social media, TikTok, vaping, young adults

## Abstract

E-cigarettes are promoted extensively on TikTok and other social media platforms. Platform policies to restrict e-cigarette promotion seem insufficient and are poorly enforced. This paper aims to understand how e-cigarettes are being promoted on TikTok and provide insights into the effectiveness of current TikTok policies. Seven popular hashtag-based keywords were used to identify TikTok accounts and associated videos related to e-cigarettes. Posts were independently coded by two trained coders. Collectively, the 264 videos received 2,470,373 views, 166,462 likes and 3426 comments. The overwhelming majority of videos (97.7%) portrayed e-cigarettes positively, and these posts received 98.7% of the total views and 98.2% of the total likes. A total of 69 posts (26.1%) clearly violated TikTok’s own content policy. The findings of the current study suggest that a variety of predominantly pro-vaping content is available on TikTok. Current policies and moderation processes appear to be insufficient in restricting the spread of pro-e-cigarette content on TikTok, putting predominantly young users at potential risk of e-cigarette use.

## 1. Introduction

E-cigarette use by young people is an issue of global public health concern [1]. The use of e-cigarettes has been described as “an epidemic among youth” by the United States Surgeon General [2], with substantial proportions of young people having tried e-cigarettes (more commonly referred to as vapes), particularly in high-income countries [3,4]. Globally, it has been estimated that 82 million people used e-cigarettes in 2021, [5]. In Australia, which has adopted a precautionary approach to e-cigarettes, recent figures show that over one-quarter (26.1%) of people aged 18–24 have tried e-cigarettes, with ever-use especially high among current smokers (63.9%) [6]. Almost three-quarters (71.9%) of young users’ report using e-cigarettes “out of curiosity”, and 21.7% use them because they believe that vaping is less harmful than regular cigarette smoking. However, evidence of the harmful health effects of e-cigarettes is growing [7], with use linked to respiratory and cardiovascular damage including airway irritation, impaired lung function, increased heart rate, and blood pressure, and aortic stiffness, as well as nicotine addiction [8,9]. 

As of 2023, the global e-cigarette market is estimated to be worth USD 24.6 billion and is predicted to increase by 3.4% over the next five years [10]. While some forms of e-cigarette advertising and promotion are regulated, such as television and radio broadcast commercials, there is mounting evidence that e-cigarette products are being promoted and advertised on social media through advertisements, social media influencers and user-generated content [11,12,13,14]. Moreover, it is often unclear if certain types of user-generated organic content, in particular influencer posts, have in fact been initiated and paid for—with money or in-kind support—by e-cigarette brands. Evidence suggests that social media messaging is dominated by positive, pro-vaping messages [11,15] which act to shape related culture and norms, and contribute to the view that e-cigarette use is common and socially accepted [16,17]. These messages may be particularly impactful on youth [13], given the high rates of youth participation in social media [18]. 

Social media platforms such as TikTok, Instagram, Facebook and YouTube are part of the fabric of everyday social life [19]. In 2021, more than 4 billion people worldwide used social media, with users spending an average of 144 min per day on social media and messaging app(lication)s [20]. TikTok is one of the fastest-growing social media platforms, consisting of short videos that can be easily shared among users [21]. In 2022, TikTok was reported to have 1.2 billion monthly active users [22] and was particularly popular among young people, with 43% of its users aged between 18 and 24 years old [23]. The vast majority (90%) of TikTok users access the app daily, with most opening the app multiple times and spending an average of 52 min per day browsing TikTok content. The content shown to TikTok users is primarily algorithm-driven, presenting users with content that is intended to reinforce their interests [16]. If a TikTok user at any point likes, views or lingers over pro-e-cigarette content, then this algorithm may have an amplifying effect, and preferentially feed users more positive e-cigarette content to their “For You” page. However, in 2022 TikTok acted on content that violates its own policies and claimed to remove content that did not meet approved content posting criteria, including videos about illegal and regulated goods, such as e-cigarettes [24]. 

Recent research has investigated the influence of exposure to e-cigarette content on social media platforms, including TikTok, on adolescents and young adults [1,16,18]. These studies have found an association between exposure to social media posts featuring e-cigarettes and increased use of e-cigarettes, lower perceptions of harms and more positive attitudes towards e-cigarettes [1]. Frequency of social media use is also important, with a study of high school students in the United States finding that those who used TikTok several times per day were more likely to initiate the use of e-cigarettes than those who used it less frequently [1]. Both e-cigarette advertising and user-generated content have been linked with a greater likelihood of initiation of e-cigarette use [18], with some noting that TikTok creators are in effect marketing e-cigarettes on behalf of the tobacco industry [16]. 

Given the high rates of youth participation in social media, including TikTok, and the evidence linking social media messaging with e-cigarette use and initiation, there is a strong case for strengthening policies around the promotion of e-cigarettes on social media. While policies to restrict content promoting tobacco products, which also include e-cigarettes, currently exist on most social media platforms, there is a clear need for tighter restrictions, moderation and enforcement [13]. These content policies are often complex, and it is unclear as to what exactly is prohibited. For example, it is not always clear whether policies only relate to direct, paid advertisements or also to other forms of promotion such as reviews, mentions and use by influencers. TikTok’s Community Guidelines policy, under the section “Drugs, controlled substances, alcohol and tobacco”, explicitly tells users: “Do not post, upload, stream or share” any content that “offers the purchase, sale, trade, or solicitation of drugs or other controlled substances, alcohol or tobacco products (including vaping products, smokeless or combustible tobacco products, synthetic nicotine products, E-cigarettes, and other ENDS [Electronic Nicotine Delivery Systems])” [25]. This includes social media influencers. 

Despite this, users are still routinely exposed to e-cigarette content on TikTok and other social media platforms [14], and it seems that current policies to restrict tobacco promotion are insufficient and confusing [13]. This paper explored how e-cigarettes are being advertised and promoted on TikTok through a content analysis of select accounts, providing an insight into the effectiveness of TikTok’s “Drugs, controlled substances, alcohol and tobacco” policy, as of February 2022. This will help to inform global public policy and practice in relation to e-cigarette promotion on social media. 

## 2. Materials and Methods

### 2.1. Data Collection

A list of popular e-cigarette-related terms was developed based on search terms from recent social media content analysis research papers [12,14,18,26,27] and trending TikTok hashtags [28]. This resulted in the following seven keywords: #juul, #iqos, #eliquid, #ecig, #disposablepod, #ejuice and #ecigarett. 

Data scraping was led by a data scientist (KC) to identify public accounts that published videos with captions containing one or more of the identified hashtags. This resulted in the identification of 2312 unique TikTok accounts. Metadata were downloaded for each account, including the total number of videos, number of relevant videos (i.e., those containing one or more of the identified hashtags), number of followers and number of likes. A subsample of the identified accounts (*n* = 115) containing the highest number of e-cigarette videos was then reviewed for eligibility purposes. In order to be included in the final sample, accounts needed to be English-speaking. A total of 14 relevant accounts were identified; however, when data were downloaded, one account had been deleted, resulting in a final sample of 13 accounts. One author (PA) reviewed each of these accounts to identify and review the relevant e-cigarette posts. The post content, including cover image, metadata, caption and video of each relevant post was downloaded in February 2022. In all, 264 posts were included for analysis.

### 2.2. Coding Frame

An inductive approach [29] informed by extant studies [12,18,27,30,31] was utilised to develop a coding framework to capture account users, views, likes, comments, shares, followers, video characteristics and content. This coding framework was tested on a random sample of 30 videos by two authors (PA and MM), with each post being viewed at least five times and then assigned codes based on the content in the video [32]. This process supported the revision of the coding framework to further refine predefined codes, merge codes and create new codes. The refined coding framework was then transferred into SPSS Statistics (v27). 

In brief, in addition to video metadata, the coding framework captured the display of e-cigarette and vape products; the sentiment of the video; the display of nicotine, addiction or health warnings; whether the post featured any promotion or product review; the display of health-related content; content specific to e-cigarette use; and other video features including animation, humour and music. More information on this coding framework is available in online Appendix A, and the dataset is available on request.

### 2.3. Analysis

Two independent coders (PA and MM) applied the modified coding framework to the sample of posts, watching each video in full and reading all captions as many times as needed to code all aspects of interest. The data were then entered into SPSS (v27). Krippendorff Alpha was conducted on the coded data to establish interrater reliability. An average score of α = 0.88 was obtained (range 0.67–1.00), indicating reasonable to perfect agreement. To optimise the final categorisation and reporting of findings, disagreements for variables were discussed and resolved between the two coders. The trained coders also reviewed the content of each video for violations against the TikTok content policy for “Drugs, controlled substances, alcohol and tobacco” [24] and identified those videos that were deemed to not meet the content policy. These identified videos were then assessed by a practising lawyer (MB) for confirmation. 

Descriptive statistics were used to summarise video metrics and characteristics. The number of likes, views, comments and shares of each post was downloaded from TikTok and used to measure popularity and engagement. The engagement rate for each video was calculated using the formula [33]:$$Engagement\;rate=\frac{\left(likes+comments+shares\right)}{views}$$

An engagement rate of 0.20 or above was used to indicate high engagement [16]. Independent samples t-tests were used to show differences in engagement rate by post characteristic. 

### 2.4. Ethical Considerations

There is consideration as to whether social media data are public or private [34]. TikTok is a data-sharing platform owned by ByteDance. With the exception of accounts owned by users who have indicated that they are aged between 13 and 15 years old when signing up to the platform, TikTok accounts are by default public, with any user having the capability to view other accounts’ videos and posts and access data using hashtags. All data collected in this study were publicly available. This study was approved by the Curtin University Human Research Ethics Committee (HR2021-0250). 

## 3. Results

The 13 accounts posted a total of 1599 videos, of which 264 (16.5%) were e-cigarette-related. The majority (*n* = 8, 61.5%) of these accounts were identified as retailers or distributors of e-cigarettes, while three were categorised as “vape enthusiasts” and two as “influencers”. Five of the accounts (38.5%) included links to an online e-cigarette retailer. The accounts were followed by a total of 183,301 users (range 4–95,100).

Collectively, the sampled e-cigarette-related posts received 2,470,373 views, 166,462 likes, 3426 comments and 1777 shares. The median number of views was 2561 (range 6–391,400), and the median number of likes was 164 (range 0–43,200). Four posts had an engagement rate greater than 0.20 (1.5%; indicating high engagement). The median engagement rate was 0.06 (range 0.00–0.25). 

The overwhelming majority of videos portrayed e-cigarette use positively (*n* = 258, 97.7%, 2,437,042 views), while only five posts portrayed e-cigarettes negatively (1.9%, 11,931 views) and one post portrayed e-cigarettes in a neutral way (0.4%, 21,400 views). Those posts portraying e-cigarette use positively received 98.7% of the total views and 98.2% of the total likes. The posts with the five highest engagement rates all portrayed e-cigarettes positively. 

Table 1 shows the number of posts featuring observed characteristics along with the associated number of views and likes. E-cigarettes were visible in the majority of posts (*n* = 208). These posts received a higher proportion of views and likes, but a significantly lower engagement rate (mean = 0.07, SD = 0.003) than those posts not showing e-cigarettes (mean = 0.09, SD = 0.008) (t(252) = −3.5, *p* < 0.001). Just over one-third of posts showed e-liquids (*n* = 90) and/or e-liquid flavours (*n* = 106). In the majority of posts (*n* = 249), the placement of e-cigarette products was overt, and the product was the clear main focus of the post. The majority of posts (*n* = 259) were pro-e-cigarettes, and only one post was anti e-cigarettes. Brands and logos were visible in 186 posts; however, only one post indicated that the post contained branded content. Over one-third of posts included a product review (*n* = 100).

While over one-quarter of posts referred to nicotine (*n* = 72), only six posts referenced vape or nicotine addiction. Four of the videos referencing addiction were pro-vaping, one was anti-vaping and one was neutral. Accordingly, a small minority of posts (*n* = 9) displayed a health warning, although none of these also referenced addiction. With regard to messaging, a small number of posts promoted e-cigarette use as a means to quit smoking (*n* = 16) or as a healthier alternative to smoking (*n* = 4). Only two posts referenced public health professionals or organisations, and four posts commented on e-cigarette regulations or policies. These posts were comparatively less popular, receiving a lower proportion of views and likes (Table 1). 

Half of the videos conveyed a shared vaping identity or community affiliation. These posts enjoyed a slightly higher mean engagement rate (mean = 0.08, SD = 0.004) than those that did not refer to a shared identity or community (mean = 0.07, SD = 0.004) (t(252) = 2.1, *p* < 0.05). 

The content of posts included references to customising vape products, accessories and juices (*n* = 29) as well as vape tricks (*n* = 47). Posts depicting vape tricks were comparatively more popular, receiving a higher proportion of views and likes (Table 1). Only one post referenced the “safe” use of e-cigarettes and related products, and three posts referenced e-cigarette devices malfunctioning. Around one-tenth of posts used animations or humour, and just over half contained background or theme music. Those posts using humour had a significantly higher mean engagement rate (mean = 0.11, SD = 0.13) than those posts which did not use humour (mean = 0.07, SD = 0.40) (t(252) = 4.6, *p* < 0.001). The majority of posts (*n* = 254) were less than 90 s in length. 

Sixty-nine posts (26.1%) were found to violate the TikTok content policy for e-cigarettes. All of these posts promoted an e-cigarette product for purchase. This included videos providing details on how and where to purchase e-cigarette products (*n* = 38), links to online retailers (*n* = 18) and links to other social media accounts for purchasing e-cigarette products (*n* = 8). Eighteen videos that were found to violate the content policy offered monetary offers such as sale prices and “buy three get one free”, and eight videos offered non-monetary offers including giveaways or free gifts with purchase. Posts violating the content policy received 407,530 views (16.5%) and 12,421 likes (7.5%) in total. These posts had a lower mean engagement rate (mean = 0.05, SD = 0.005) than those posts which did not violate the content policy (mean = 0.08, SD = 0.003) (t(252) = −4.5, *p* < 0.001).

## 4. Discussion

Our study selected 13 English-language TikTok accounts containing e-cigarette videos in order to explore how e-cigarettes are promoted on the platform, and whether current platform content policies are effective. The overwhelming majority of posts portrayed e-cigarettes in a positive light, consistent with previous research looking at e-cigarette posts on various social media platforms [15,27,35], including TikTok [16,18]. In addition, in most posts, e-cigarettes were visible and shown overtly, with the e-cigarette product being the main focus of the post. 

The use of certain characteristics, such as shared identity, humour and vaping tricks in social media posts may act to shape social norms around e-cigarette use and increase the perception that use is normal and socially accepted [18]. Half of the posts sampled in the current study made reference to a vaping community or shared identity through the use of specific hashtags. This is in line with a previous study of e-cigarette content on Instagram, which found that posts frequently referred to vaping community hashtags [36], as well as research with Australian vapers who interacted with the online international e-cigarette network to discuss personal experiences, acquire relevant skills and troubleshoot techniques [37]. People who identify with pro-vaping communities are more likely to hold negative attitudes towards quitting e-cigarettes and exhibit lower behavioural control over e-cigarette use [38]. 

Just under one-fifth of the posts sampled in the current study showed or referenced vaping tricks, similar to past findings [18]. Posts showing vaping tricks were also seen to be comparatively more popular, attracting a greater proportion of views and likes. Early research has shown that adolescents identify vaping tricks as a reason for their initiation of e-cigarette use [39,40], although tricks may be declining in importance, with a 2022 study showing tricks to be of very low importance among adolescent e-cigarette users in Australia [41]. That study found that flavours and taste were the most important characteristics of e-cigarettes to adolescents, in line with the current study’s findings that 40% of videos showed e-liquid flavours. Humour was not often used in the posts sampled here, in contrast to previous research findings that comedy was a prominent theme in TikTok videos addressing e-cigarette use [14,18]. The use of humour may be an effective tool to reach young audiences, with fun and comedy being core motives for social media users [18]. Accordingly, our study found that those videos which included humour had a higher engagement score. 

Consistent with a previous content analysis of e-cigarette videos on TikTok [18], health warnings were not commonly seen in the sampled posts. Similarly, an analysis of e-cigarette-related tweets found that only a minority included health warnings or mentioned age restrictions [27]. In addition, a recent focus group study found that adolescents expressed concerns about the lack of health and age warnings on social media posts around e-cigarette use [42]. There was also an absence of public health professionals and critical voices in the posts sampled here, with only two videos referencing a public health professional. There is a clear need for evidence-based challenges to the dominant pro-vaping messages present on social media platforms, with public health professionals being a key source of this anti-vaping content [15]. 

Despite TikTok content policy expressly prohibiting the promotion of tobacco and e-cigarette products for sale, around one-quarter of posts were seen to promote the purchase of vape products and over one-third included a product review. Encouragingly, these posts were less popular, and users were less likely to engage with these posts than with non-promotional posts. Previous studies have also found that product promotion is a common feature of TikTok videos [14,18] and Twitter posts [12]. Many posts also expressly showed e-cigarette brands and logos, although only one post indicated that the post was sponsored, using the hashtag #promotions. It is likely that at least some of these posts violated TikTok policy, which states that branded content that promotes e-cigarette products should not be posted [25]. 

In total, 69 posts were found to violate the TikTok content policy, consistent with findings that the violation of social media platform policies is common [13]. This highlights the shortcoming of relying on platforms to develop and enforce their own social media content guidelines, as despite influencer industry guidelines and platform policies demanding transparency and disclosure, in reality, in-kind and paid promotions and product placement are frequently not clearly labelled and divulged, leaving it up to audiences to read between the lines. There is an evident need for policy wording to be modified to capture more posts promoting e-cigarettes, and for more emphasis to be placed on implementing these policies, including through moderation. Further, there are no significant consequences for those who do not follow TikTok’s guidelines and violate content policy [43]. Social media platforms can decide the consequences for breaches of their policies, but they have a clear financial incentive not to punish people who breach their policies. Government regulation that captures e-cigarette advertising, promotion and sponsorship, including on social media, is needed. This must include requiring social media platforms to report on how they are ensuring these regulations are upheld. The WHO Framework Convention on Tobacco Control has encouraged all parties to include e-cigarettes in existing comprehensive bans on tobacco advertising [44].

The current study was limited in that the sampled posts were relatively small in number and restricted to English-language videos. However, this study was exploratory in nature, and the sample size here was similar to other studies exploring the promotion of e-cigarettes on social media platforms including TikTok [16,26] (*n* = 148 and 100) and Instagram [35] (*n* = 85). In addition, the sample did not capture posts that may have been related to e-cigarettes but did not use the identified hashtags. However, we used the most popular e-cigarette terms based on recent research [12,14,18,26,27] and trending TikTok hashtags [28] at the time of data collection. The data were collected around the time of TikTok’s content policy changes (February 2022), and it is unknown whether some of the posts we captured here may have been subsequently removed through moderation policies. We were unable to download metadata for nine of the posts, and eight hashtags that we had initially considered during the hashtag identification process (#vapelife, #vapetricks, #vapecommunity, #vapenation, #vapeshop, #vapefam, #vapeporn and #vapeon) had been removed from TikTok at the time of data collection. 

We used engagement rate, calculated as the number of likes, comments and shares divided by the number of views, to assess user engagement with videos. We acknowledge that this method does not take into account how long a video has been posted, and also does not consider those accounts where these features have been disabled. However, this method has been used in recent research [16,45], and we have also directly reported the number of likes and views received. Finally, we are unable to determine the effects of exposure to the posts studied here, although past research suggests that the content of these posts is likely to lead to the normalisation of e-cigarette use, particularly among young people [17]. 

## 5. Conclusions

Together, the findings of this study suggest that a variety of predominantly pro-vaping information is being communicated and viewed on TikTok. Much of this content violates current social media platform policies, and it is clear that these policies and current moderation methods are insufficient to restrict the spread of promotional e-cigarette content. There is a need for evidence-informed and engaging anti-vaping content on platforms such as TikTok to counter or moderate the pro-vaping content. Greater government regulation of e-cigarette promotions is also needed to ensure that social media platforms such as TikTok do not promote these harmful products, particularly to young people.

## Figures and Tables

**Table 1 ijerph-20-05761-t001:** Observed characteristics, views and likes of e-cigarette-related posts on TikTok (*n* = 264).

	N	%	Views	%	Likes	%
	264		2,470,373		166,462	
Products						
E-cigarettes visible	208	78.8	2,379,377	96.3	158,364	95.1
E-liquid/e-juice products visible	90	34.1	357,247	14.5	28,303	17.0
E-liquid flavours visible or mentioned	106	40.2	546,369	22.1	24,141	14.5
Product shown overtly and as main focus of post	249	94.3	2,419,479	97.9	160,887	96.7
Brand or logo visible or mentioned	186	70.5	1,776,157	71.9	109,990	66.1
Sentiment						
Pro-vaping	259	98.1	2,461,452	99.6	165,843	99.6
Anti-vaping	1	0.4	1850	0.1	125	0.1
Type						
Product review	100	37.9	719,629	29.1	39,474	23.7
Promotes purchase of vape product	69	26.1	407,530	16.5	12,421	7.5
Content						
Displays/refers to nicotine content	72	27.3	356,607	14.4	14,015	8.4
References nicotine or vape addiction	6	2.3	35,544	1.4	3169	1.9
Displays/refers to health warning	9	3.4	22,529	0.9	1590	1.0
Promotes vaping as a means to quit smoking	16	6.1	63,645	2.6	5555	3.3
Promotes vaping as a healthier alternative to smoking	4	1.5	26,220	1.1	1827	1.1
References public health professionals or organisations	2	0.8	2804	0.1	156	0.1
Comments on e-cigarette regulation or policy	4	1.5	4668	0.2	284	0.2
Customisation of e-cigarette products and/or juices	29	11.0	197,329	8.0	8988	5.4
Vape tricks	47	17.8	914,018	37.0	51,460	30.9
References “safe” use of e-cigarettes	1	0.4	2426	0.1	263	0.2
References e-cigarette devices malfunctioning	3	1.1	12,669	0.5	1391	0.8
Video features and characteristics						
Conveys shared vaping identity or community affiliation	132	50.0	972,433	39.4	79,591	47.8
Contains animations or cartoons	27	10.2	24,334	1.0	1138	0.7
Uses humour	26	9.8	88,172	3.6	8849	5.3
Contains background or theme music	147	55.7	1,469,428	59.5	84,689	50.9
Video length						
90 s or less in duration	254	96.2	2,329,214	94.3	160,528	96.4
More than 90 s in duration	10	3.8	141,159	5.7	5934	3.6

## Data Availability

Data available on application.

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
