# Peer review of "Promotion of E-Cigarettes on TikTok and Regulatory Considerations"

_ijerph, 2023, doi:10.3390/ijerph20105761_

Round 1
Reviewer 1 Report
Thank you for the opportunity to review the article “Promotion of e-cigarettes on TikTok and regulatory considerations”. The paper addresses an interesting and well researched theme in the recent period about the implications, challenges and difficulties of defining limits and policies of promotion goods and/ or services via Social Media platforms. The paper is submitted under the section “Health Behavior, Chronic Disease and Health Promotion”, special issue “Public Health Impacts of Exposure to Tobacco, Alcohol, and Other Substances Content on Social Media”.
This study represents a new approach in the field, discussing the subject that needs to be comprehensively analyzed because “while some forms of e-cigarette advertising and promotion are regulated, such as television and radio broadcast commercials, there is mounting evidence that e-cigarette products are being promoted and advertised on social media through advertisements, social media influencers, and user-generated content”, as the authors underline.
Also, the study is written in an adequate manner and the results are presented clearly and coherently, using visuals and text. The table presented in the paper was relevant to explore the results of the proposed article and the ways this was adapted to the explanations in the text.
As the authors say, one conclusion assumes that this article points “that a variety of predominantly pro-vaping information is being communicated and viewed on TikTok. Much of this content violates current social media platform policies, and it is clear that these policies and current moderation methods are insufficient to restrict the spread of promotional e-cigarette content”.
Moreover, there are some observations that should be addressed in this revision.
- Line 58-59, the idea needs proper citation.
- Line 74 “Recent research”, please add several citations to support the idea.
- In section Materials and Methods please state very clearly the methodology involved, as the authors describe it as an exploratory study (Line 118), a case study (103), but also using content analysis (Line 127).
- The “selected 13 English language TikTok accounts” (Line 121, Line 228) are all from Australia? Is there a connection with the partial declared objective at Line 105 “This will help to inform Australian”. If so, please revise the title to add this specific information.
- Please add an Annex with all the 13 English language TikTok accounts.
Author Response
Comment 1.
Thank you for the opportunity to review the article “Promotion of e-cigarettes on TikTok and regulatory considerations”. The paper addresses an interesting and well researched theme in the recent period about the implications, challenges and difficulties of defining limits and policies of promotion goods and/ or services via Social Media platforms. The paper is submitted under the section “Health Behavior, Chronic Disease and Health Promotion”, special issue “Public Health Impacts of Exposure to Tobacco, Alcohol, and Other Substances Content on Social Media”.
This study represents a new approach in the field, discussing the subject that needs to be comprehensively analyzed because “while some forms of e-cigarette advertising and promotion are regulated, such as television and radio broadcast commercials, there is mounting evidence that e-cigarette products are being promoted and advertised on social media through advertisements, social media influencers, and user-generated content”, as the authors underline.
Also, the study is written in an adequate manner and the results are presented clearly and coherently, using visuals and text. The table presented in the paper was relevant to explore the results of the proposed article and the ways this was adapted to the explanations in the text.
As the authors say, one conclusion assumes that this article points “that a variety of predominantly pro-vaping information is being communicated and viewed on TikTok. Much of this content violates current social media platform policies, and it is clear that these policies and current moderation methods are insufficient to restrict the spread of promotional e-cigarette content”.
Response 1
We thank the reviewer for their thoughtful comments. We have addressed further comments below.
Comment 2.
Line 58-59, the idea needs proper citation.
Response 2
We have added the following reference to this sentence.
Pagoto, S.; Waring, M.E.; Xu, R. A call for a public health agenda for social media research. J Med Internet Res, 2019. 21(12): e16661. DOI: 10.2196/16661
Comment 3.
Line 74 “Recent research”, please add several citations to support the idea.
Response 3
We have added the following references to this sentence.
Vassey, J.; Galimov, A.; Kennedy, C.J.; Vogel, E.A.; Unger, J.B. Frequency of social media use and exposure to tobacco or nicotine-related content in association with E-cigarette use among youth: A cross-sectional and longitudinal survey analysis. Prev Med Rep, 2022. 30: 102055. DOI: 10.1016/j.pmedr.2022.102055
Morales, M.; Fahrion, A.; Watkins, S.L. #NicotineAddictionCheck: Puff Bar culture, addiction apathy, and promotion of e-cigarettes on TikTok. Int J Environ Res Public Health, 2022. 19(3). DOI: 10.3390/ijerph19031820
Sun, T.; Lim, C.C.W.; Chung, J.; Cheng, B.; Davidson, L.; Tisdale, C.; Leung, J.; Gartner, C.E.; Connor, J.; Hall, W.D.; Chan, G.C.K. Vaping on TikTok: A systematic thematic analysis. Tobacco control, 2021: tobaccocontrol-2021-056619. DOI: 10.1136/tobaccocontrol-2021-056619.
Comment 4.
In section Materials and Methods please state very clearly the methodology involved, as the authors describe it as an exploratory study (Line 118), a case study (103), but also using content analysis (Line 127).
Response 4
We now refer to this study as a content analysis and have deleted the other terms. The statement now reads:
“This paper explored how e-cigarettes are being advertised and promoted on TikTok through a content analysis of select accounts…”
Comment 5.
The “selected 13 English language TikTok accounts” (Line 121, Line 228) are all from Australia? Is there a connection with the partial declared objective at Line 105 “This will help to inform Australian”? If so, please revise the title to add this specific information.
Response 5
The TikTok accounts are not Australian (five are from the United Kingdom, five from the United States, and three could not be determined). We have removed “Australian” from the objective in Lines 105-106, so that it now reads:
“This will help to inform global public policy and practice in relation to e-cigarette promotion on social media.”
Comment 6.
Please add an Annex with all the 13 English language TikTok accounts.
Response 6
Unfortunately, our ethics application does not allow us to name TikTok accounts. We are therefore unable to fulfill this request.
Reviewer 2 Report
This manuscript reported a content analysis of 264 TikTok videos related to e-cigarettes. The authors aimed to understand how e-cigarettes are being promoted on TikTok. The research topic is socially significant and relevant to the readers of the journal. The rationale and discussion were well-written. The overall research design seemed valid. I commend the authors’ hard work and contribution to the literature. However, I have several questions and comments, which I elaborate on below.
1. The introduction section was well-written. The authors provided a strong justification for studying e-cigarette videos on TikTok.
2. Lines 118-120, Page 3: Please clarify how the subsample of 115 accounts were selected. The authors wrote, they “containing the greatest number of e-cigarette videos.” What does this mean? Does this mean all the 115 accounts have the exact same number of e-cigarette videos? What is this number?
3. In the Coding Frame section, I’d suggest the authors briefly mention the coding categories of this study, that is, what had been coded. The current writing requires readers to refer to the supplemental table, which may confuse and burden readers.
4. I don’t think “Genre” is a good label for the last coding category. Genre refers to different categories of videos. I think “Video feature/characteristics” may be a more accurate label to describe what had been coded here.
5. When it comes to the “Genre” category, the information mentioned in the text (Lines 210-214, Page 5), table 1, and supplemental table seemed inconsistent. For instance, the supplemental table mentioned “Identity or community,” which was not discussed in the text or presented in the table. In addition, the table 1 did not mention “humor,” but used “attempts to provoke laughter and provide amusement.” This may confuse readers. Lastly, the “Video 90 seconds or less in duration” and “Video more than 90 seconds in duration” in Table 1 can be combined, as they are the same variable.
6. In my humble opinion, the engagement rate formula may not be valid. First of all, this formula was not published in a peer-reviewed source. Second, how long a video has been posted likely affects the numbers of likes, comments, shares, and views in different ways/rates. Thus, the current formula does not consider the impact of when/how long a video has been posted on user engagement. Third, some TikTok accounts may disable some of the functions such as commenting and sharing. Thus, the engagement rate formula needs to be used with caution. I’d strongly suggest the authors use the numbers of likes, comments, shares, and views directly as an additional way to access user engagement, and explicitly acknowledge the limitation of using these indicators.
7. The authors did a great job interpreting their results and providing practical suggestions.
Author Response
Comment 1.
This manuscript reported a content analysis of 264 TikTok videos related to e-cigarettes. The authors aimed to understand how e-cigarettes are being promoted on TikTok. The research topic is socially significant and relevant to the readers of the journal. The rationale and discussion were well-written. The overall research design seemed valid. I commend the authors’ hard work and contribution to the literature. However, I have several questions and comments, which I elaborate on below.
Response 1
We thank the reviewer for their thoughtful comments. We have addressed further comments below.
Comment 2.
The introduction section was well-written. The authors provided a strong justification for studying e-cigarette videos on TikTok.
Response 2
We thank the reviewer for this kind comment.
Comment 3.
Lines 118-120, Page 3: Please clarify how the subsample of 115 accounts were selected. The authors wrote, they “containing the greatest number of e-cigarette videos.” What does this mean? Does this mean all the 115 accounts have the exact same number of e-cigarette videos? What is this number?
Response 3
Of the accounts that were identified through the keyword search, we selected those accounts which had the highest number of e-cigarette related videos. We have changed the wording in Lines 120-121 to clarify this. These accounts did not all have the same number of e-cigarette videos.
“a subsample of the identified accounts (n=115) containing the highest number of e-cigarette videos were then reviewed for eligibility purposes”
Comment 4.
In the Coding Frame section, I’d suggest the authors briefly mention the coding categories of this study, that is, what had been coded. The current writing requires readers to refer to the supplemental table, which may confuse and burden readers.
Response 4
We have added some text around the main categories of the coding framework to the “Coding Frame” section in Lines 138-142, as follows:
“In brief, in addition to video metadata, the coding framework captured the display of vape products; the sentiment of the video; the display of nicotine, addiction, or health warnings; whether the post featured any promotion or product review; the display of health-related content; content specific to e-cigarette use; and other video features including animation, humour, and music. More information on this coding framework is available in online supplemental table S1, and the dataset is available on request.”
Comment 5.
I don’t think “Genre” is a good label for the last coding category. Genre refers to different categories of videos. I think “Video feature/characteristics” may be a more accurate label to describe what had been coded here.
Response 5
We have changed the label of this coding category to “Video features and characteristics.”
Comment 6.
When it comes to the “Genre” category, the information mentioned in the text (Lines 210-214, Page 5), table 1, and supplemental table seemed inconsistent. For instance, the supplemental table mentioned “Identity or community,” which was not discussed in the text or presented in the table. In addition, the table 1 did not mention “humor,” but used “attempts to provoke laughter and provide amusement.” This may confuse readers. Lastly, the “Video 90 seconds or less in duration” and “Video more than 90 seconds in duration” in Table 1 can be combined, as they are the same variable.
Response 6
We have revised the text and tables to ensure that they are consistent, as follows:
- Rows in Table 1 have been rearranged to more closely resemble supplementary table S2.
- We have revised the heading “Conveys shared vaping community affiliation” to “Conveys shared vaping identity and community affiliation” in Table 1 to match the wording in supplementary table S2 more closely. Accordingly, we have added the word “identity” to the text in Line 210 where identity and community affiliation are discussed.
- The heading “Attempts to provoke laughter and provide amusement” in Table 1 has been changed to “Uses humour”.
- The video length variable in Table 1 has been combined.
Comment 7.
In my humble opinion, the engagement rate formula may not be valid. First of all, this formula was not published in a peer-reviewed source. Second, how long a video has been posted likely affects the numbers of likes, comments, shares, and views in different ways/rates. Thus, the current formula does not consider the impact of when/how long a video has been posted on user engagement. Third, some TikTok accounts may disable some of the functions such as commenting and sharing. Thus, the engagement rate formula needs to be used with caution. I’d strongly suggest the authors use the numbers of likes, comments, shares, and views directly as an additional way to access user engagement, and explicitly acknowledge the limitation of using these indicators.
Response 7
We acknowledge the reviewer’s reservations about the engagement rate formula. We believe the engagement rate formula, which has been used in previous peer-reviewed literature (e.g. Morales et al, 2022, IJERPH; Winzer et al, 2022, IJERPH) adds insights into the data. Therefore we would like it to remain in the manuscript. However, we have acknowledged the limitations of using these metrics to assess engagement in the Discussion (Lines 318-323).
“We used engagement rate, calculated as the number of likes, comments, and shares divided by the number of views, to assess user engagement with videos. We acknowledge that this method does not take into account how long a video has been posted, and also does not consider those accounts where these features have been disabled. However, this method has been used in recent research [16, 45], and we have also directly reported the number of likes and views received..”
Comment 8.
The authors did a great job interpreting their results and providing practical suggestions.
Response 8
We thank the reviewer for this comment.
Round 2
Reviewer 1 Report
Thank you for the opportunity to review the revised version of the paper “Promotion of e-cigarettes on TikTok and regulatory considerations”. The paper was submitted under the section “Health Behavior, Chronic Disease and Health Promotion”, special issue “Public Health Impacts of Exposure to Tobacco, Alcohol, and Other Substances Content on Social Media”.
The authors responded and explained with reasonable arguments and corrected all the remarks and observations highlighted in the previous review and the results suggest a more consistent and logical text.
To sum it up, the authors developed a more in-depth theoretical presentation about the subject, integrating the suggested aspects of the review.
I consider that the paper is publishable after a final check from the authors.
Reviewer 2 Report
The authors have sufficiently addressed my questions and concerns. I have no further comments. I appreciate the authors' hard work and contribution to the literature.